# Characterization of the literature informing health care of transgender and gender-diverse persons: A bibliometric analysis

Badal S. B. Pattar[1,2,3], Nabilah Gulamhusein[1,2,3], Chantal L. Rytz[1,2,3], Keila Turino Miranda[4], Lauren B. Beach[5], Zack Marshall[1,3], David Collister[6], Dina N. Greene[7], Cameron T. Whitley[8], Nathalie Saad[1,9], Sandra M. Dumanski[1,2,3,9], Tyrone G. Harrison[1,2,3,9], Lindsay Peace[10], Amelia M. Newbert[10], Sofia B. Ahmed[1,6,11] *

1 Cumming School of Medicine, University of Calgary, Calgary, Alberta, Canada, 2 Libin Cardiovascular Institute, University of Calgary, Calgary, Alberta, Canada, 3 O'Brien Institute for Public Health, University of Calgary, Calgary, Alberta, Canada, 4 Cardiovascular Health and Autonomic Regulation Laboratory, Department of Kinesiology and Physical Education, McGill University, Montreal, Quebec, Canada, 5 Department of Medical Social Sciences, Northwestern Feinberg School of Medicine, Chicago, Illinois, Unites States of America, 6 Department of Medicine, University of Alberta, Edmonton, Alberta, Canada, 7 Department of Laboratory Medicine, University of Washington, Seattle, Washington, Unites States of America, 8 Department of Sociology, Western Washington University, Bellingham, Washington, United States of America, 9 Department of Medicine, University of Calgary, Calgary, Alberta, Canada, 10 Skipping Stone Foundation, Calgary, Alberta, Canada, 11 Women and Children's Health Research Institute, University of Alberta, Edmonton, Alberta, Canada

* Sofia.Ahmed@albertahealthservices.ca

## Abstract

### Background and objective

Transgender and gender-diverse (TGD) persons experience health inequities compared to their cisgender peers, which is in part related to limited evidence informing their care. Thus, we aimed to describe the literature informing care provision of TGD individuals.

### Data source, eligibility criteria, and synthesis methods

Literature cited by the World Professional Association of Transgender Health Standards of Care Version 8 was reviewed. Original research articles, excluding systematic reviews (n = 74), were assessed (n = 1809). Studies where the population of interest were only caregivers, providers, siblings, partners, or children of TGD individuals were excluded (n = 7). Results were synthesized in a descriptive manner.

### Results

Of 1809 citations, 696 studies met the inclusion criteria. TGD-only populations were represented in 65% of studies. White (38%) participants and young adults (18 to 29 years old, 64%) were the most well-represented study populations. Almost half of studies (45%) were cross-sectional, and approximately a third were longitudinal in nature (37%). Overall, the median number of TGD participants (median [IQR]: 104 [32, 356]) included in each study was approximately one third of included cisgender participants (271 [47, 15405]). In studies

**Data Availability Statement:** All relevant data are within the manuscript and its Supporting information files.

**Funding:** The author(s) received no specific funding for this work.

**Competing interests:** The authors have declared that no competing interests exist.

where both TGD and cisgender individuals were included (n = 74), the proportion of TGD to cisgender participants was 1:2 [1:20, 1:1]. Less than a third of studies stratified results by sex (32%) or gender (28%), and even fewer included sex (4%) or gender (3%) as a covariate in the analysis. The proportion of studies with populations including both TGD and cisgender participants increased between 1969 and 2023, while the proportion of studies with study populations of unspecified gender identity decreased over the same time period.

## Conclusions

While TGD participant-only studies make up most of the literature informing care of this population, longitudinal studies including a diversity of TGD individuals across life stages are required to improve the quality of evidence.

## Introduction

The United States Preventive Services Task Force (USPSTF) has recently recognized the limited evidence informing the preventive care of populations based on gender identity; a catalog of current USPSTF suggestions revealed insufficient evidence to make any specific recommendations for transgender and gender diverse (TGD) populations [1]. Underscoring the critical nature of literature informing the optimal care of TGD persons, a 2023 national cohort study found that TGD individuals in a universal health care system have a risk of all-cause mortality that is between 1.34 and 1.75 times greater than that of cisgender people [2].

The vital importance of diverse representation in study participants to optimize health outcomes has been previously highlighted [3], with calls for greater inclusion of participants from gender minorities in clinical research [4, 5]. A recent report from the National Academies of Sciences, Engineering, and Medicine [6] underscores the impact of inequitable representation, leading to compromised generalizability of clinical research findings, increased health system costs, and lack of access to effective medical interventions; all of these complications compound intersectional health disparities in the populations currently underrepresented or excluded in clinical trials and research. Historically, there has been insufficient representation of women and minoritized populations in research, but representation has steadily increased over the last several decades at least in part due to changes in policies [7, 8] and law [9].

In 2017, ClinicalTrials.gov introduced an optional gender eligibility description data field [10]. While only 25% of entries contained gender identity terms with a range of frequencies [11], this represents a promising step towards increasing TGD representation in research. Overall, there has been increasing recognition of sex and gender as key variables in research, and the importance of representation of diverse populations in health research. Thus, we conducted a bibliometric analysis of the World Professional Association of Transgender Health Standards of Care Version 8 (WPATH SOC 8) [12], a comprehensive guideline based on the best available science and expert professional consensus for the health of TGD individuals. Our primary objective was to describe the participant demographics and study design of the literature in WPATH SOC 8 informing the care of TGD persons. Our secondary objective was to specifically describe the literature informing gender-affirming hormone therapy guidelines, given recent heightened focus on this area of TGD healthcare.

## Material and methods

### Data source and searches

The reference list from WPATH SOC 8 was used as a proxy for influential literature related to care of TGD persons, which consists of 18 chapters that each focus on a specific aspect of TGD health care. All citations from WPATH SOC 8 were manually exported from each chapter and organized into 18 groups in EndNote (Version 9.3.3, Clarivate, London, United Kingdom).

### Study selection

All published primary research articles were included. Systematic reviews (n = 74) were excluded to avoid "double-counting participants", as were narrative reviews (n = 232) commentaries (n = 103), editorials (n = 30). Studies where the population of interest were only caregivers, providers, siblings, and partners (n = 6), or children of TGD individuals (n = 1) were also excluded. Fig 1 includes the complete list of exclusion reasons. Articles in languages other than English (n = 5) were translated to English using Google Translate [13], which has been previously validated in systematic reviews [14]. Two reviewers (BSBP, NG) independently extracted data using a standardized data abstraction form in the Covidence platform (Cochrane Technology, Melbourne, Victoria, Australia). Extracted study characteristics included the publication year, the chapter(s) in which the reference was cited, study design, gender (e.g., TGD, cisgender, TGD and cisgender, not reported, etc.), race/ethnicity, and age composition of study population, sample size, proportion of TGD to cisgender individuals if these populations were included in the study, whether the final results were stratified by sex and/or gender, and whether sex and/or gender were included as a covariate in the study analysis. Data was also extracted on whether studies reported disability and neurodivergent status of participants [15, 16]. A quality assurance exercise was employed following extraction, where a random sample of 10% of all included articles were assessed in duplicate for consistency. Agreement between the two reviewers (BSBP, NG) was calculated by dividing the number of articles where the abstracted data was identical between reviewers by the total number of articles assessed. Abstracted data was identical in 70% of articles and 30% of articles had at least one discrepancy, which were resolved by a third reviewer (SBA).

### Statistical analysis

Results were synthesized in a descriptive manner. Data were presented as medians and range for the entirety of the WPATH SOC 8, as well as by chapter, including the literature informing Chapter 12 "Hormone Therapy". The linear-by-linear trend test [17] was used to determine temporal patterns in the gender identity of included participants and study designs, where a significant change in trend was indicated by $p<0.05$. All analyses were performed using Stata (Version 18, StataCorp, College, Texas, United States of America).

## Results

Of 1809 cited articles, 696 studies met the inclusion criteria (Fig 1). The median year of publication was 2018 (range: 1969 to 2023) (Table 1). Although WPATH SOC 8 was published in 2022, 3 included articles were listed as "in press" and later published in 2023. TGD-only populations were represented in 65% of studies (Table 1), with young adults (18 to 29 years old, 64%) being the most well-represented group. Race/ethnicity was not reported in 59% of studies. Of the studies reporting race/ethnicity, white participants were the most represented (38%) (median per study [interquartile range (IQR)]) (130 [38, 640]), followed by Black or African descent (29%, 13 [3, 167]), Hispanic/Latinx (25%, 43 [6, 246]), and Asian/Pacific

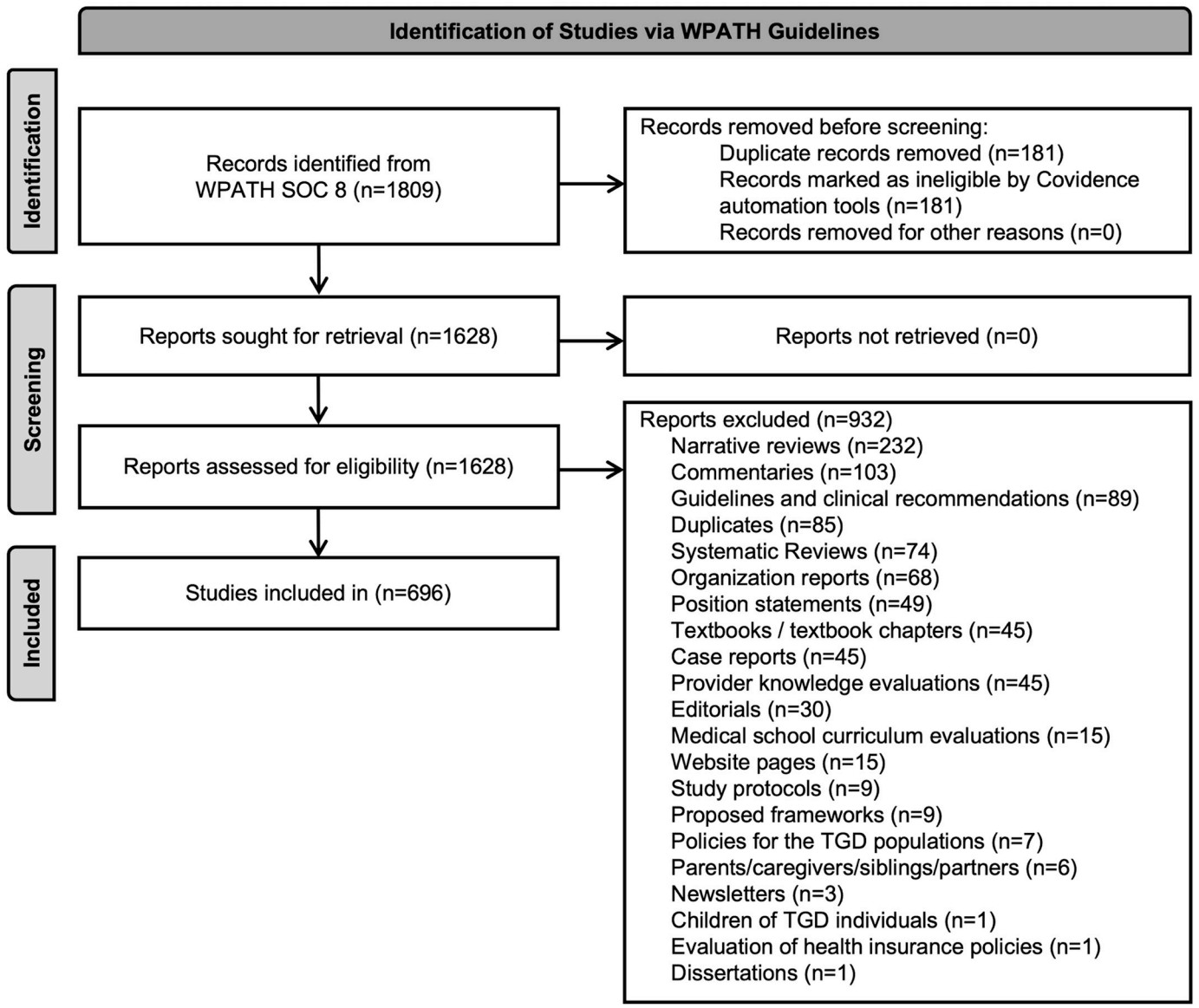

**Fig 1. Preferred reporting items for systematic reviews and meta-analyses (PRISMA) 2020 flow of studies.** TGD: Transgender and Gender-Diverse. WPATH SOC 8: World Professional Association of Transgender Health Standards of Care Version 8.

Islander (22%, 9 [3, 77]) participants. The most common study design was cross-sectional (45%), followed by prospective cohort (17%), retrospective cohort (14%), and qualitative (13%). Longitudinal studies accounted for approximately a third of study designs (37%). Overall, the number of TGD participants per study (median [interquartile range (IQR)]) (104 [32, 356]) was approximately one third of cisgender participants (271 [47, 15405]). Less than one third of studies stratified results by sex (32%) or gender (28%) and even fewer included sex (4%) or gender (3%) as a covariate in analyses. A minority of studies included disability (22%) or neurodivergent (5%) status of participants. In studies where both TGD and cisgender individuals were included (n = 74), the median proportion of TGD to cisgender participants was 1:2 [1:20, 1:1]. Data for individual chapters, excluding Chapter 12, are listed in S1 File.

**Table 1. Summary of study characteristics of studies across all 18 chapters.**

| | All | Cross Sectional | Prospective Cohort | Retrospective Cohort | Qualitative | Mixed-Methods | Randomized Controlled Trial | Case Control | Pre-Post |
|---|---|---|---|---|---|---|---|---|---|
| | n = 696 (100%) | n = 312 (45%) | n = 120 (17%) | n = 92 (14%) | n = 91 (13%) | n = 39 (5%) | n = 19 (3%) | n = 13 (2%) | n = 10 (1%) |
| Median Year (Range) | 2018 (1969–2023) | 2018 (1999–2023) | 2017 (1969–2022) | 2018 (1997–2022) | 2018 (2008–2023) | 2016 (2000–2022) | 2019 (2003–2021) | 2018 (2006–2021) | 2016 (1997–2020) |
| **Gender Identities n (%)** | | | | | | | | | |
| Cisgender | 9 (1) | 1 (<1) | 2 (2) | 1 (1) | - | - | 2 (11) | 3 (23) | - |
| Eunuch | 6 (1) | 4 (1) | - | - | 1 (1) | 1 (3) | - | - | - |
| Intersex | 7 (1) | 2 (<1) | 1 (<1) | - | 2 (2) | 2 (5) | - | - | - |
| LGBTQ | 27 (4) | 12 (4) | - | 1 (1) | 9 (10) | 4 (10) | - | - | 1 (10) |
| Not Reported | 80 (11) | 32 (10) | 22 (18) | 7 (8) | 10 (11) | 2 (5) | 5 (26) | - | 2 (20) |
| Questioning | 43 (6) | 16 (5) | 18 (15) | 2 (2) | 3 (3) | 2 (5) | 1 (5) | 1 (8) | - |
| TGD + Cisgender | 74 (11) | 46 (15) | 8 (7) | 7 (8) | 4 (4) | 3 (8) | - | 6 (46) | - |
| TGD | 450 (65) | 199 (64) | 69 (58) | 74 (80) | 62 (68) | 25 (64) | 11 (58) | 3 (23) | 7 (70) |
| **Race / Ethnicity n (%) \*** | | | | | | | | | |
| Asian or Pacific Islander | 151 (22) | 86 (28) | 16 (13) | 11 (12) | 22 (24) | 8 (21) | 4 (21) | 2 (15) | 2 (20) |
| Black or African Descent | 200 (29) | 117 (38) | 19 (16) | 14 (15) | 31 (34) | 9 (23) | 5 (26) | 3 (23) | 2 (20) |
| Hispanic or Latinx | 172 (25) | 106 (34) | 15 (13) | 11 (12) | 25 (27) | 7 (18) | 4 (21) | 2 (15) | 2 (20) |
| Indigenous or Aboriginal | 80 (11) | 54 (17) | 8 (7) | - | 9 (10) | 4 (10) | 2 (11) | 1 (8) | 2 (20) |
| Multiple Races/Ethnicities | 119 (17) | 67 (21) | 11 (10) | 3 (3) | 27 (30) | 6 (39) | 2 (11) | 1 (8) | 2 (20) |
| Not Disclosed or Unknown | 103 (15) | 57 (18) | 13 (11) | 10 (11) | 8 (9) | 6 (15) | 6 (32) | 3 (23) | - |
| Not Reported | 409 (59) | 158 (51) | 85 (71) | 73 (79) | 41 (45) | 26 (67) | 10 (53) | 9 (69) | 7 (70) |
| Other | 153 (22) | 100 (32) | 17 (14) | 12 (13) | 12 (13) | 3 (8) | 6 (32) | 2 (15) | 1 (10) |
| White | 264 (38) | 144 (46) | 32 (27) | 17 (18) | 43 (47) | 13 (33) | 8 (42) | 4 (31) | 3 (30) |
| **Age Groups Represented n (Years; %) \*** | | | | | | | | | |
| 12 ≤ age | 91 (13) | 37 (12) | 28 (23) | 10 (11) | 9 (10) | 3 (8) | 2 (11) | 1 (8) | 1 (10) |
| 13 ≤ age ≤ 17 | 193 (28) | 94 (30) | 36 (30) | 22 (24) | 21 (23) | 9 (23) | 3 (16) | 2 (15) | 6 (60) |
| 18 ≤ age ≤ 29 | 446 (64) | 201 (64) | 75 (63) | 57 (62) | 63 (69) | 28 (72) | 11 (58) | 3 (23) | 8 (80) |
| 30 ≤ age ≤ 50 | 374 (54) | 168 (54) | 60 (50) | 54 (59) | 49 (54) | 23 (59) | 12 (63) | 5 (38) | 3 (30) |
| 51 ≥ age | 263 (38) | 132 (42) | 35 (29) | 35 (38) | 34 (38) | 14 (36) | 6 (32) | 4 (31) | 2 (20) |
| **Stratification by Gender? n (%)** | | | | | | | | | |
| Yes | 195 (28) | 110 (35) | 32 (27) | 23 (25) | 4 (4) | 9 (23) | 4 (21) | 9 (69) | 3 (30) |
| **Gender Included as a Covariate in Analysis? n (%)** | | | | | | | | | |
| Yes | 24 (3) | 13 (4) | 8 (7) | - | 1 (1) | - | 1 (5) | 1 (8) | - |
| **Stratification by Sex? n (%)** | | | | | | | | | |
| Yes | 224 (32) | 128 (41) | 40 (33) | 22 (24) | 6 (7) | 10 (26) | 5 (26) | 7 (54) | 5 (50) |
| **Sex Included as a Covariate in Analysis? n (%)** | | | | | | | | | |
| Yes | 25 (4) | 16 (5) | 6 (5) | 1 (1) | 1 (1) | 1 (3) | - | - | - |
| **Reported Disability? n (%)** | | | | | | | | | |
| Yes | 156 (22) | 81 (26) | 19 (16) | 25 (27) | 10 (11) | 6 (15) | 6 (32) | 6 (46) | 3 (30) |
| **Reported Neurodiversity? n (%)** | | | | | | | | | |

*(Continued)*

**Table 1.** (Continued)

|  | All | Cross Sectional | Prospective Cohort | Retrospective Cohort | Qualitative | Mixed-Methods | Randomized Controlled Trial | Case Control | Pre-Post |
|---|---|---|---|---|---|---|---|---|---|
|  | **n = 696 (100%)** | **n = 312 (45%)** | **n = 120 (17%)** | **n = 92 (14%)** | **n = 91 (13%)** | **n = 39 (5%)** | **n = 19 (3%)** | **n = 13 (2%)** | **n = 10 (1%)** |
| Yes | 36 (5) | 16 (5) | 9 (8) | 5 (5) | 3 (3) | 1 (3) | - | 2 (15) | - |

Notes. White: minimum value (%). Darkest shade of grey: maximum value (%). Values in between are indicated by the gradient. LGBTQ: Lesbian, Queer, Bisexual, Transgender, and Queer. Questioning: Individuals who were questioning their gender identity. TGD: Transgender and Gender-Diverse.

**\*:** Percentages add up to more than 100% and n adds up to more than total number of studies due to multiple age groups and/or racial/ethnic groups being represented in some studies.

## Temporal patterns in the gender composition of study populations and study designs

Despite the gender composition of studies published yearly since 1969 remaining stable (Fig 2A), there has been significant increase (p = 0.03) in the proportion of studies involving both TGD and cisgender individuals and a significant decrease in the proportion of studies involving study participants with an unspecified gender identity (p = 0.04; Fig 2B). The proportions TGD-only (p = 0.36) or cisgender-only studies (p = 0.82) did not change over the same time period.

From 1969 to 2023, we observed a change in the composition of study designs. Longitudinal studies were initially most common, with a temporal increase in cross sectional, qualitative and mixed method studies (Fig 3A). Moreover, the proportion of longitudinal studies decreased (p = 0.04) while the proportion of qualitative studies increased (p = 0.03; Fig 3B). The proportions of cross sectional (p = .09) or mixed-methods (p = .80) did not change over the same time period.

## Study characteristics informing hormone therapy guidance

Of 1809 cited articles, 115 original studies met the inclusion criteria (Table 2). The median year of publication was 2018 (range: 1981 to 2022). TGD-only populations were represented in 72% of studies, with 18 to 29 years of age being the most represented age group across all studies (63%), followed by 30 to 50 years of age (58%). Race/ethnicity was not reported in 57% of studies. Of the studies that did report race/ethnicity, white participants had the highest representation (42%), followed by Black or African descent (30%), Asian or Pacific Islander (28%), and Hispanic/ Latinx (26%) participants. The most common study design was cross-sectional (34%), with randomized controlled trials (RCTs) representing 8% of studies. Less than half of studies stratified results by sex (48%) or gender (39%), and only a minority included sex (2%) or gender (2%) as a covariate in relevant analyses. Approximately a third of studies reported disability status (32%), and 6% reported neurodiversity status.

## Discussion

We sought to characterize the literature directly informing care of TGD persons. Our key findings were as follows: 1) two-thirds of studies included TGD-only populations, 2) the proportion of studies with populations that include both TGD and cisgender participants has increased over time, while the proportion of studies with study populations of unspecified gender identity has decreased over time, 3) the majority of studies included cross-sectional studies within young adults age ranges, 4) while the majority of studies did not report the race/

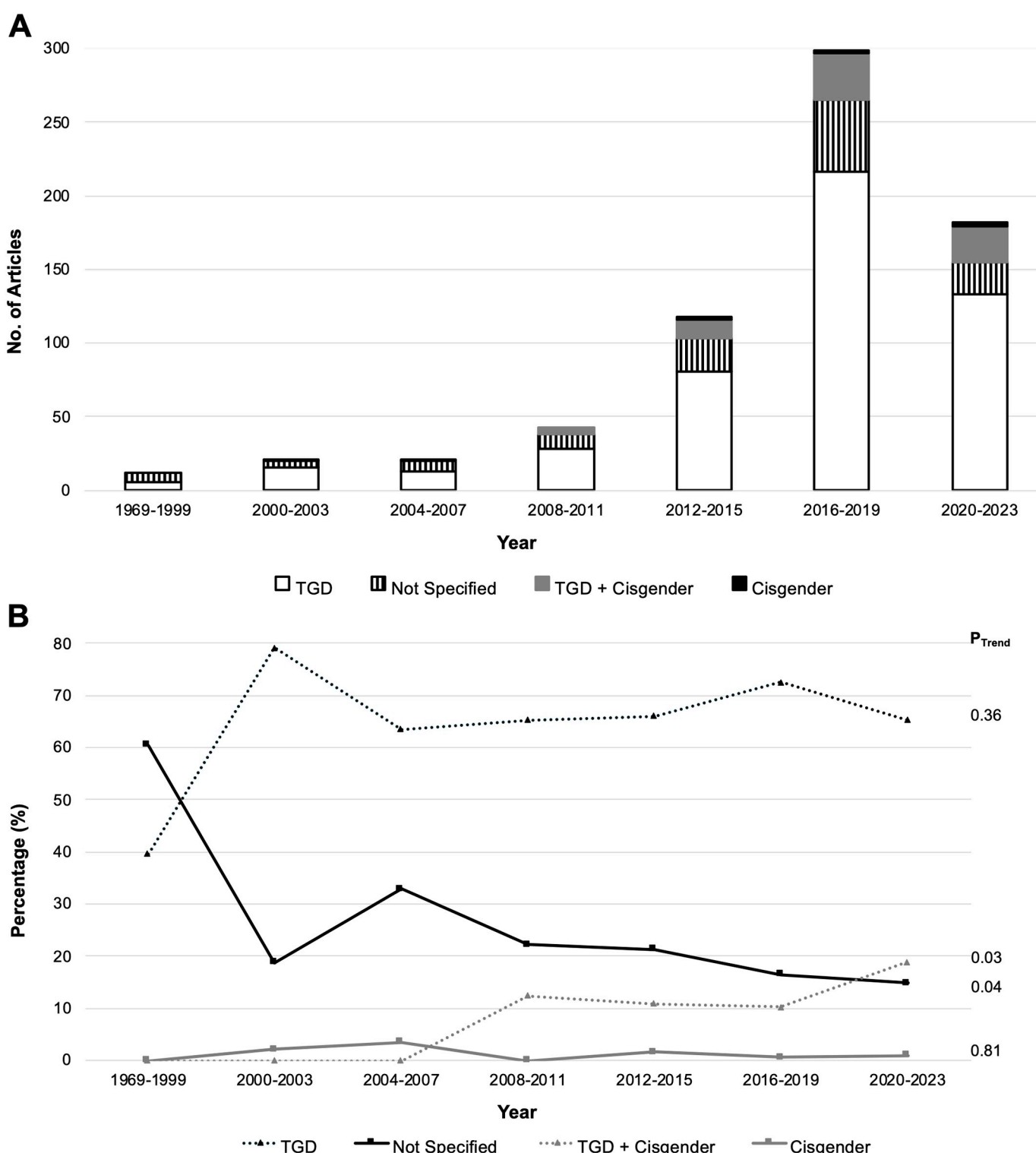

**Fig 2. Temporal patterns in the gender composition of study populations among studies cited in the WPATH SOC 8.** A) Absolute number of studies published and gender identity of participants as a function of time. B) Gender identity of participants involved in studies over time expressed as percentage. $P_{Trend}$ calculated for the proportion of studies involving exclusively TGD individuals, not specifying gender identity, both TGD and cisgender individuals, and exclusively cisgender individuals. $P_{Trend}$: p-value indicating if there is a significant change in trend over the time period. TGD: Transgender and Gender-diverse. Given terms including eunuch, intersex, LGBTQ, do not provide sufficient information to categorize an individual's gender identity, they were included in the "Not Specified" group. Participants who were questioning their gender identity were included in the "TGD" group.

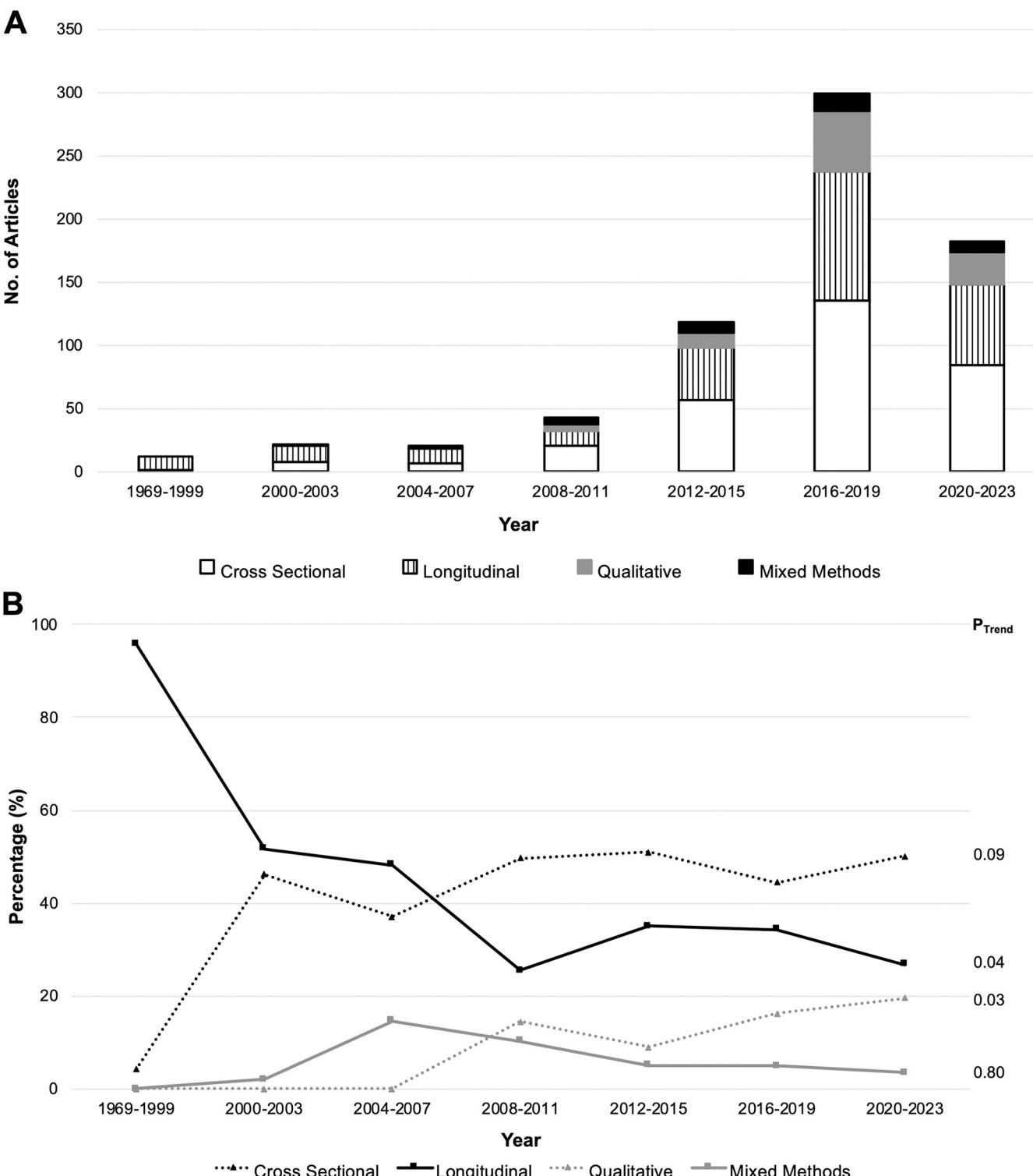

**Fig 3. Temporal patterns in study designs among studies cited in the WPATH SOC 8.** A) Absolute number of studies published and study designs as a function of time. B) Design of studies over time expressed as percentage. $P_{Trend}$ calculated for the proportion of studies involving cross sectional, longitudinal, qualitative, and mixed-methods studies. $P_{Trend}$: p-value indicating if there is a significant change in trend over the time period. Longitudinal studies consisted of prospective and retrospective cohort studies, randomized controlled trials, case control, and pre-post studies.

**Table 2. Summary of study characteristics of studies cited in chapter 12, titled "Hormone Therapy".**

| | All | Cross Sectional | Prospective Cohort | Retrospective Cohort | Qualitative | Mixed-Methods | Randomized Controlled Trial | Case Control | Pre-Post |
|---|---|---|---|---|---|---|---|---|---|
| | n = 115 (100%) | n = 39 (34%) | n = 29 (25%) | n = 19 (17%) | n = 4 (3%) | n = 8 (7%) | n = 9 (8%) | n = 5 (4%) | n = 2 (2%) |
| Median Year (Range) | 2018 (1981–2022) | 2017 (2000–2021) | 2018 (1981–2022) | 2019 (1997–2021) | 2018 (2015–2020) | 2014 (2008–2021) | 2013 (2003–2021) | 2014 (2007–2019) | 2019.5 (2019–2020) |
| **Gender Identities n (%)** | | | | | | | | | |
| Cisgender | 9 (8) | 1 (3) | 2 (7) | 1 (5) | - | - | 2 (22) | 3 (60) | - |
| Intersex | - | - | - | - | - | - | - | - | - |
| Eunuch | - | - | - | - | - | - | - | - | - |
| LGBTQ | 2 (2) | - | - | - | 1 (25) | 1 (12) | - | - | - |
| Not Reported | 4 (3) | 2 (5) | 2 (7) | - | - | - | - | - | - |
| Questioning | 8 (7) | 1 (3) | 5 (17) | 1 (5) | - | 1 (12) | - | - | - |
| TGD + Cisgender | 9 (8) | 5 (13) | 2 (7) | - | - | 1 (12) | - | 1 (20) | - |
| TGD | 83 (72) | 30 (77) | 18 (62) | 17 (90) | 3 (75) | 5 (63) | 7 (78) | 1 (20) | 2 (100) |
| **Race / Ethnicity n (%) *** | | | | | | | | | |
| Asian or Pacific Islander | 32 (28) | 14 (36) | 4 (14) | 5 (26) | 2 (50) | 2 (25) | 3 (33) | 1 (20) | 1 (50) |
| Black or African Descent | 35 (30) | 16 (41) | 5 (17) | 4 (21) | 2 (50) | 2 (25) | 4 (44) | 1 (20) | 1 (50) |
| Hispanic or Latinx | 30 (26) | 13 (33) | 4 (14) | 5 (26) | 2 (50) | 2 (25) | 3 (33) | - | 1 (50) |
| Indigenous or Aboriginal | 17 (15) | 9 (23) | 3 (10) | - | 1 (25) | 1 (13) | 2 (22) | - | 1 (50) |
| Multiple Races/Ethnicities | 19 (17) | 11 (28) | 2 (7) | 1 (5) | 3 (75) | 1 (13) | - | - | 1 (50) |
| Not Disclosed or Unknown | 28 (24) | 10 (26) | 5 (17) | 6 (32) | - | 2 (29) | 4 (44) | 1 (20) | - |
| Not reported | 65 (57) | 19 (49) | 20 (69) | 13 (68) | - | 5 (63) | 4 (44) | 3 (60) | 1 (50) |
| Other | 26 (23) | 13 (33) | 5 (17) | 3 (16) | 1 (25) | - | 3 (33) | 1 (20) | - |
| White | 48 (42) | 19 (49) | 8 (28) | 6 (32) | 4 (100) | 3 (38) | 5 (56) | 2 (40) | 1 (50) |
| **Age Groups Represented n (Years; %) *** | | | | | | | | | |
| 12 ≤ age | 11 (10) | 3 (8) | 5 (13) | 3 (16) | - | - | - | - | - |
| 13 ≤ age ≤ 17 | 30 (26) | 11 (28) | 8 (21) | 6 (32) | 1 (25) | 2 (25) | - | 1 (20) | 1 (50) |
| 18 ≤ age ≤ 29 | 72 (63) | 23 (59) | 19 (49) | 13 (68) | 3 (75) | 5 (63) | 6 (67) | 2 (40) | 1 (50) |
| 30 ≤ age ≤ 50 | 67 (58) | 21 (54) | 15 (38) | 12 (63) | 2 (50) | 3 (38) | 9 (100) | 5 (100) | - |
| 51 ≥ age | 41 (36) | 16 (41) | 6 (15) | 7 (37) | 2 (50) | 1 (13) | 5 (56) | 4 (80) | - |
| **Stratification by Gender? n (%)** | | | | | | | | | |
| Yes | 45 (39) | 12 (31) | 15 (52) | 10 (53) | - | 3 (38) | 2 (22) | 2 (40) | 1 (50) |
| **Gender Included as a Covariate in Analysis? n (%)** | | | | | | | | | |
| Yes | 2 (2) | 1 (3) | - | - | - | - | - | 1 (20) | - |
| **Stratification by Sex? n (%)** | | | | | | | | | |
| Yes | 55 (48) | 17 (44) | 18 (62) | 10 (34) | - | 3 (38) | 3 (33) | 2 (40) | 2 (100) |
| **Sex Included as a Covariate in Analysis? n (%)** | | | | | | | | | |
| Yes | 2 (2) | 2 (5) | - | - | - | - | - | - | - |
| **Reported Disability? n (%)** | | | | | | | | | |
| Yes | 37 (32) | 14 (36) | 6 (21) | 6 (32) | - | 4 (50) | 4 (44) | 3 (67) | - |
| **Reported Neurodiversity? n (%)** | | | | | | | | | |

*(Continued)*

**Table 2.** (Continued)

| | All | Cross Sectional | Prospective Cohort | Retrospective Cohort | Qualitative | Mixed-Methods | Randomized Controlled Trial | Case Control | Pre-Post |
|---|---|---|---|---|---|---|---|---|---|
| | n = 115 (100%) | n = 39 (34%) | n = 29 (25%) | n = 19 (17%) | n = 4 (3%) | n = 8 (7%) | n = 9 (8%) | n = 5 (4%) | n = 2 (2%) |
| Yes | 7 (6) | 4 (10) | 3 (10) | - | - | - | - | - | - |

Notes. White: minimum value (%). Darkest shade of grey: maximum value (%). Values in between are indicated by the gradient. LGBTQ: Lesbian, Queer, Bisexual, Transgender, and Queer. Questioning: Individuals who were questioning their gender identity. TGD: Transgender and Gender-Diverse.

**\*:** Percentages add up to more than 100% and n adds up to more than total number of studies due to multiple age groups and/or racial/ethnic groups being represented in some studies.

ethnicity of participants, white participants were most commonly represented in studies listing this demographic, and 5) hormone therapy guidelines are largely informed by studies including TGD participants. These results suggest improved methods with greater and more diverse participant representation within TGD-associated health research leading to an enriched evidence base informing the care of TGD persons. However, much work still remains in both TGD-associated health research [18], and health research at large [1, 11].

Our results are consistent with previous reports also showing a shift towards more TGD-centric health research, although we were not able to determine if this shift was inclusive of all or only limited to specific gender identities. In a bibliometric analysis of peer-reviewed literature in TGD health between 1900 and 2017, an exponential increase in the number of TGD health articles was observed over the last decade [19]. However, as outlined previously [18, 20, 21], more comprehensive long-term research centering a diversity of TGD health experiences across life stages is required to optimally inform care of persons of different gender identities. In particular, guidance related to gender-affirming hormone therapy has been informed by the principles of hormone replacement treatment of hypogonadal individuals [22], which may or may not reflect the needs of TGD individuals. Previous work has outlined the perils of generalizing results from one population to other populations [3, 23–25]; our analysis show that in addition to a much larger absolute number of cisgender participants compared to TGD participants across all studies in all chapters, nine studies included cisgender participants exclusively, and these nine studies were all used to inform gender-affirming hormone therapy in TGD individuals.

The percentage of older adults (individuals aged ≥65 years [26] is projected to double from 12% to 22% by 2050 [27] due to declining fertility and increasing longevity [28]. Although numerous studies included participants from across the lifespan, young adults were the most frequently captured age group, which may reflect a primary research focus on time periods just before or early into the gender transition process [18], thus limiting the potential for informed care of TGD individuals in the decades after medical transition or older adults initiating the gender transition process. Underrepresentation of older adults in clinical research is widespread across virtually all fields [29–35] with resulting care guidelines often relying on studies with limited older participants, potentially reducing their applicability in this age group [36–38]. Furthermore, very few studies reported participant disability or neurodiversity status. Previous studies have suggested increased rates of neurodevelopmental diagnoses in TGD individuals relative to cisgender individuals [39, 40], and thus appropriate representation in study participants is crucial to tailor care meeting the needs of TGD populations. A recent review [41] outlined common barriers and potential solutions to increase participation of

older adults in clinical research, which coupled with TGD-inclusive practices [42, 43], could lead to a greater evidence base in the health care of TGD persons.

It is noteworthy that cross-sectional studies were the most common type of study informing care of TGD persons and that this study type is not designed to determine causal relationships [44]. Two recent reviews exploring TGD research also found cross-sectional studies to be the most common type of study [21, 45]. As outlined above, there is a need for longitudinal studies examining TGD health throughout the lifespan [18, 21]. However, the significant increase in the proportion of qualitative studies is encouraging as it may reflect a shift towards a greater number of studies examining TGD individuals' experiences [46]. The quality of evidence from health research is partially deemed by the hierarchy of study designs, where observational studies are considered of lower quality evidence compared to RCTs, and systematic reviews and meta-analyses. As previously highlighted, increased scientific rigor and reach of TGD health research is required to ultimately inform evidence-based prevention and care for this underserved population [47]. Of note, RCTs are not always appropriate or ethical, particularly in the absence of clinical equipoise, and observational studies may be more reflective of "real world" data [48].

It is encouraging that our observations show an increase in the proportion of studies including both TGD and cisgender participants, in parallel to a decrease in the proportion of studies with study populations of unspecified gender identity. These observations are likely due to a myriad of factors, including changes in research methods and approaches to data collection and policy [10] as well as increased availability of educational resources, guidelines, and frameworks [49–52]. However, in studies where both TGD and cisgender participants were included, for every one TGD individual there were two cisgender individuals, and the median number of cisgender participants was approximately triple that of the median number of TGD participants included across all studies. Previous studies have highlighted the importance of the participation to prevalence ratio (PPR), a measure of the representation of a particular group in the study population relative to the prevalence of the condition of interest in the same group [53]. While we did not stratify studies according to condition, study populations including both TGD and cisgender participants should be representative of the prevalence of the condition under examination. The low proportions of studies incorporating sex and gender identity in analysis of results are in keeping with previous studies [20, 21, 54]. At a time when TGD health research is under scrutiny, and in alignment with journal [50], funding agencies [55–57] and policy framework [58] guidance on the inclusion of sex and gender in research, prospectively planning for both inclusion and disaggregation of sex and gender is required to support sex- and gender-based analyses of not only TGD health research, but all health research.

This study has limitations. First, only studies referenced in the WPATH SOC 8 were included, rather than the entire body of literature informing TGD healthcare. However, WPATH is an international and multidisciplinary professional association whose mission is to promote evidence-based care of TGD persons; one of the main functions of WPATH is to promote the highest standards of health care for TGD persons through the Standards of Care [12] "based on the best available science". While important literature may have been missed, the methods used to develop WPATH SOC 8 are publicly available [58] and align with existing frameworks for the development of clinical practice guidelines [59, 60]. Next, studies where the population of interest assessing caregivers, providers, siblings, partners, or children of TGD individuals were excluded, thus centering the focus on only TGD individuals, which does not capture issues in the broader environment or family systems that play important roles in health outcomes [18]. However, as the aim of this work as to specifically examine TGD representation in health research informing the care of TGD persons, inclusion of these other

groups would not have contributed to this goal. It is also important to note the potential bias toward studies with more TGD populations. It should not be unexpected that a data set including studies informing the care of the TGD population have a high proportion of TGD study participants. However, literature used to inform guidelines has been previously shown to not be inclusive of sex and gender, as well as other intersectional, considerations [54, 61–66]. Articles in languages other than English were translated to English using Google Translate [13] rather than having a native language speaker review the studies, potentially impacting the accuracy of our review. Nevertheless, a previous study has validated the use of this translation tool in systematic reviews, demonstrating an 85–97% accuracy when compared to native language speakers [14]. Finally, WPATH SOC 8 was published in 2022, and as such new scientific evidence published since its release was not captured in this study. In February 2024, a literature search on Pubmed.gov using the keyword "transgender" and "2023" revealed 134 RCTs, systematic reviews and meta-analyses, which may reflect an even greater evidence base for the care of TGD individuals.

## Conclusions

This review was aimed to characterize the literature informing the care of TGD persons as outlined in the WPATH SOC version 8 guidelines. While it is encouraging to observe that TGD participant-only studies represent the majority of this literature, the quality of evidence is an area of concern and greater attention to inclusion of a diversity of TGD individuals across the lifespan and especially as TGD individuals age, in addition to community engagement in all steps of the research process, is warranted [18, 20, 21, 42, 45]. In this era of precision health [67], the growing number of individuals who identify as TGD [68–70], coupled with the increasingly recognized health inequities within this understudied and underserved population [2, 71, 72], underscores the need for urgent action.

## Supporting information

**S1 File. Data for individual chapters, excluding Chapter 12.**
(DOCX)

**S2 File. Studies and study characteristics included in bibliometric analysis.**
(XLSX)

## Author Contributions

**Conceptualization:** Badal S. B. Pattar, Chantal L. Rytz, Keila Turino Miranda, Lindsay Peace, Sofia B. Ahmed.

**Data curation:** Badal S. B. Pattar, Nabilah Gulamhusein, Sofia B. Ahmed.

**Formal analysis:** Badal S. B. Pattar, Chantal L. Rytz, Lauren B. Beach, Sofia B. Ahmed.

**Investigation:** Badal S. B. Pattar, Nabilah Gulamhusein.

**Methodology:** Badal S. B. Pattar, Nabilah Gulamhusein, Chantal L. Rytz, Keila Turino Miranda, Lauren B. Beach, Zack Marshall, David Collister, Dina N. Greene, Cameron T. Whitley, Nathalie Saad, Sandra M. Dumanski, Tyrone G. Harrison, Lindsay Peace, Sofia B. Ahmed.

**Project administration:** Badal S. B. Pattar.

**Supervision:** Sofia B. Ahmed.

**Validation:** Badal S. B. Pattar.

**Visualization:** Badal S. B. Pattar, Chantal L. Rytz, Sofia B. Ahmed.

**Writing – original draft:** Badal S. B. Pattar, Sofia B. Ahmed.

**Writing – review & editing:** Badal S. B. Pattar, Nabilah Gulamhusein, Chantal L. Rytz, Keila Turino Miranda, Lauren B. Beach, Zack Marshall, David Collister, Dina N. Greene, Cameron T. Whitley, Nathalie Saad, Sandra M. Dumanski, Tyrone G. Harrison, Lindsay Peace, Amelia M. Newbert, Sofia B. Ahmed.

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
