## [Decision Letter · Decision Letter 0]

29 May 2024

PONE-D-24-13832Characterization of the literature informing health Care of transgender and gender-diverse persons: A bibliometric analysisPLOS ONE

Dear Dr. Ahmed,

Thank you for submitting your manuscript to PLOS ONE. After careful consideration, we feel that it has merit but does not fully meet PLOS ONE’s publication criteria as it currently stands. Therefore, we invite you to submit a revised version of the manuscript that addresses the points raised during the review process.

Thank you submitting your manuscript for review. The peer reviewers have suggest adding clarifications or elaborations regarding several points in your manuscript. I think addressing these concerns will improve the clarity of your manuscript and utility to a wide audience. I share the reviewer’s concerns about Figure 2 in your manuscript. What information does this figure convey that is not already presented in Table 1? If you want to retain the figure, I suggest making it more apparent what additional information is being presented Reviewer 1 has provided feedback below and as notes in a pdf of you manuscript (see attached). Please respond to both sets of recommendations. Christina Roberts, MD MPH

We look forward to receiving your revised manuscript.

Kind regards,

Christina M. Roberts, M.D., M.P.H.

Academic Editor

PLOS ONE

Reviewers' comments:

Reviewer's Responses to Questions

**Comments to the Author**

1. Is the manuscript technically sound, and do the data support the conclusions?

Reviewer #1: Yes

Reviewer #2: Partly

2. Has the statistical analysis been performed appropriately and rigorously? 

Reviewer #1: Yes

Reviewer #2: Yes

3. Have the authors made all data underlying the findings in their manuscript fully available?

Reviewer #1: Yes

Reviewer #2: Yes

4. Is the manuscript presented in an intelligible fashion and written in standard English?

Reviewer #1: Yes

Reviewer #2: Yes

5. Review Comments to the Author

Reviewer #1: I really appreciate this unbiased analysis of the literature from which WPATH have drawn their latest standards of care.

I have a number of suggested revisions to maximise its utility to the published literature.

Some of these are minor in nature in a described in comments in the PDF attached.

More broadly, I think the authors would do well to expand the analysis in order to draw stronger conclusions.

WPATH recommendations have a number of restrictions on how they are drawn up, including the need to be globally applicable. Although broader literature can be cited in the text, the restrictions on the recommendations, along with word count, may limit the citing of 'state of the art research'. This is worth acknowledging.

For this reason, it may be useful to compare SOC8 studies longitudinally but to the previous SOC7.

Further, the authors have a specific endpoint around studies related hormone therapy. It would be helpful to contrast this sub analysis to the remainder of the literature (is it more or less robust)? And also to the Endocrine Society guideline references, as the authors explicitly state these have over-reliance on studies with cisgender participants only.

Although these do constitute significant further analysis they would add weight to conclusions on the direction of the field and point to a need for more frequent guideline review.

At a time when literature on TGD health is under scrutiny, the authors may also wish to acknowledge this and suggest pragmatic ways forward that best serve the needs of the population.

Reviewer #2: PONE-D-24-13832 Reviewer Comments

Bibliographic review – see Donthu et al 2021

Overall impressions

This is a reasonable paper with appropriate methods to answer the questions posed in the introduction. A bibliometric analysis is well conducted (though I am not an expert in this method). The conclusions are a bit broad for the data set used (Literature supporting TGD care) as it over-reaches in its claim that these results possibly demonstrate an increase in inclusion of TDG people in health research in general, which is implied in the conclusion. This was an analysis of the TGD literature in a guideline that was focused on best practices for TGD care, so it is expected to contain primary studies with this population and is not a very novel finding. It does make good points about the type of literature and demographic gaps in the knowledge base that will inform future studies. The analysis of the hormonal care section is sound, though there is a missed opportunity to include analysis of other sections in a similar manner.

Specific Points

Introduction

Clear rational for the study, given the underrepresented transgender populations and gender in general in scientific studies.

79 The secondary objective is vague in the introduction, with no indication of what they want to explore about gender-affirming hormone therapy guidelines. It is clear later in the article that they are applying the same bibliometric analysis to hormone therapy guidelines, so this should be explicit in the objectives. Also, they presented data only on hormone therapy guidance, not surgical or other care, which are an important part of the guidelines. If this was the only portion of the guidelines that was important it should be explicitly stated in the objectives and give us a reason why surgical guidelines, fertility preservation, sexual and mental health etc. were not included.

Methods

74 A bibliometric review appears to be a straightforward investigation method to answer the study question about trends in gender based research.

89 Inclusion criteria is appropriate, and the authors should be commended on including non-English studies by sending them through Google Translate. It would be helpful to see a reference that details the accuracy of Google Translate for scientific studies.

96 Extraction characteristics appear appropriate for a bibliometric analysis.

104 With straight forward data extraction, and little subjectivity, I would expect the agreement between reviewers to be higher. There is no mention of inter-rater reliability statistic and how it was generated for the quality assurance portion of the review, though I like the thoroughness of including this QA. Consider including more detail about how this was calculated or if it was ad hoc.

110 Linear-by-linear trend test appears to be an appropriate statistic to uncover trends in the temporal data.

Results

Well reported, clearly written, tables are well laid out and easy to interpret.

Figure 1 has spelling error “sough” appears to be “sought”.

To make it clear that the exclusion criteria were very specific and broad, adding a note in methods to see figure 1 for complete list of exclusion criteria would be helpful, there were many more categories than expected based on the methods described in the methods section.

Figure 2 is clear, but does not add much beyond the table, just a re-representation of the data in visual format. Authors could consider leaving this out, unless the editors feel it further clarifies the findings in table 1.

Figure 3: Temporal patterns graphic is very pixilated, this may be the packaging software that collated the document, if not then a higher res graphic would be needed. Cannot interpret the hashed sections effectively and had to guess.

148-153 It is not unexpected that a data set that included primary literature that was informing the care of TGD populations to have a higher percentage of TGD participants. See note below.

157 PTrend is not defined in the fig 3 caption.

Discussion

188 The conclusion that these results “suggest a change towards improved inclusion of methods allowing for greater representation of a diversity of participants within health research” is not supported by this study. This study looked at TGD associated research only, I suspect that TGD populations are still very unrepresented in health research in general, and this study did not look beyond the literature informing TDG care. Suggest a revision to narrow the scope of this conclusion.

198 The secondary outcome focusing on hormone therapy seems to be a more important finding, in that the guidance is somewhat based on hypogonadal (cisgender?) individuals.

207-219 Excellent observation re: age representation and need for research that captures longitudinal data, this is one of the major findings and informs future research design.

248 Limitations are well laid out. The above limitation and possibility of bias toward studies with more TGD populations may be included here.

---

## [Author Response · Author response to Decision Letter 0]

8 Jul 2024

July 8, 2024

Emily Chenette, PhD 

Editor-in-Chief 

PLOS ONE

Manuscript ID#: PONE-D-24-13832.R1

Title: Characterization of the literature informing health care of transgender and gender-diverse persons: A bibliometric analysis

Thank you for allowing us the opportunity to resubmit our manuscript, based on the thorough review and thoughtful comments provided by the Reviewers. We feel that we have been able to satisfactorily address all the comments raised, which has further enhanced the quality of the manuscript. We have provided an itemized summary of the changes made to the paper below with the Reviewers’ comments provided in bold, followed by our responses. We have provided a tracked changed version of the manuscript, as well as a version without track changes.

Editor:

C1. I share the reviewer’s concerns about Figure 2 in your manuscript. What information does this figure convey that is not already presented in Table 1? If you want to retain the figure, I suggest making it more apparent what additional information is being presented.

R1. We thank the Editor for this comment. Figure 2 has been removed from the manuscript.

C2. Please ensure that your manuscript meets PLOS ONE's style requirements, including those for file naming. The PLOS ONE style templates can be found at

R2. We thank the Editor for this comment. The manuscript has now been formatted to meets PLOS One's style requirements. 

C3. We note that the grant information you provided in the ‘Funding Information’ and ‘Financial Disclosure’ sections do not match.

R3. Thank you for this comment. This study was unfunded, and the ‘Funding Information’ section has been updated accordingly.

C4. When completing the data availability statement of the submission form, you indicated that you will make your data available on acceptance. We strongly recommend all authors decide on a data sharing plan before acceptance, as the process can be lengthy and hold up publication timelines. Please note that, though access restrictions are acceptable now, your entire data will need to be made freely accessible if your manuscript is accepted for publication. This policy applies to all data except where public deposition would breach compliance with the protocol approved by your research ethics board. If you are unable to adhere to our open data policy, please kindly revise your statement to explain your reasoning and we will seek the editor's input on an exemption. Please be assured that, once you have provided your new statement, the assessment of your exemption will not hold up the peer review process.

R4. We thank the Editor for this comment. We have submitted a supplementary file, which includes our entire dataset to generate all figures and tables found in our manuscript. We have also listed this file under the “Supporting Information” header one page 22, line 317:

“S2 File. Studies and study characteristics included in bibliometric analysis.”

C5. We note that you have included the phrase “data not shown” in your manuscript. Unfortunately, this does not meet our data sharing requirements. PLOS does not permit references to inaccessible data. We require that authors provide all relevant data within the paper, Supporting Information files, or in an acceptable, public repository. Please add a citation to support this phrase or upload the data that corresponds with these findings to a stable repository (such as Figshare or Dryad) and provide and URLs, DOIs, or accession numbers that may be used to access these data. Or, if the data are not a core part of the research being presented in your study, we ask that you remove the phrase that refers to these data.

R5. We thank the Editor for this comment. The phrase “data not shown” has been removed. We have now indicated that these results are reported in Table 1 and 2, as well as the supplementary file. 

C6. Please review your reference list to ensure that it is complete and correct. If you have cited papers that have been retracted, please include the rationale for doing so in the manuscript text, or remove these references and replace them with relevant current references. Any changes to the reference list should be mentioned in the rebuttal letter that accompanies your revised manuscript. If you need to cite a retracted article, indicate the article’s retracted status in the References list and also include a citation and full reference for the retraction notice.

We thank the Editor for the comment. The following references have been updated: 

Line 327, citation #4: National Academies of Sciences, Engineering and Medicine. Measuring Sex, Gender Identity, and Sexual Orientation. Washington (DC): National Academies Press (US). 2022.

Line 333, citation #6: National Academies of Sciences, Engineering and Medicine. Improving Representation in Clinical Trials and Research: Building Research Equity for Women and Underrepresented Groups. Washington (DC): National Academies Press (US). 2022.

Line 337, citation #7: National Institutes of Health Research: Office of Research on Women’s Health. NIH Policy on Sex as a Biological Variable 2015 [Available from: https://orwh.od.nih.gov/sex-gender/orwh-mission-area-sex-gender-in-research/nih-policy-on-sex-as-biological-variable].

Line 341, citation #8: National Institutes of Health Research. Amendment: NIH Policy and Guidelines on the Inclusion of Women and Minorities as Subjects in Clinical Research 2017 [Available from: https://grants.nih.gov/grants/guide/notice-files/NOT-OD-18-014.html].

Line 344, citation #9: National Institutes of Health Research. NIH Revitalization Act of 1993. Public Law. 1993:103-43.

Line 395, citation #28: United Nations. Ageing [Available from: https://www.un.org/en/global-issues/ageing#:~:text=While%20declining%20fertility%20and%20increasing,in%20some%20countries%20and%20regions].

Line 447, citation #45: National Institutes of Health Research. The Sexual & Gender Minority Research Office 2015 [Available from: https://dpcpsi.nih.gov/sgmro].

Line 498, citation #63: Hudson K., Lifton, R., Patrick-Lake, B., Burchard. E.G., Coles, T., Collins. R., et al. The Precision Medicine Initiative Cohort Program – Building a Research Foundation for 21st Century Medicine 2015 [Available from: https://acd.od.nih.gov/documents/reports/DRAFT-PMI-WG-Report-9-11-2015-508.pdf].

The following references have been added to address the Reviewers’ constructive comments:

Line 356, citation #14: Jackson JL, Kuriyama A, Anton A, Choi A, Fournier J-P, Geier A-K, et al. The Accuracy of Google Translate for Abstracting Data From Non–English-Language Trials for Systematic Reviews. Annals of Internal Medicine. 2019;171(9):677-9.

Line 359, citation #15: World Health Organization. International classification of functioning, disability and health : ICF. Geneva: World Health Organization; 2001.

Line 361, citation #16: Chapman R. Defining neurodiversity for research and practice. 2020. p. 218-20.

Line 465, citation #51: Canadian Institute of Health Research. How to integrate sex and gender into research 2019 [Available from: https://cihr-irsc.gc.ca/e/50836.html].

Line 467, citation #52: National Institutes of Health Research: Center for Scientific Review. Premise, Rigor, Sex as a Biological Variable 2020 [Available from: https://public.csr.nih.gov/FAQs/ReviewersFAQs/PremiseRigorSexBiologicalVariable#7].

Line 470, citation #53: European Commission. Gender in EU research and innovation 2023 [Available from: https://rea.ec.europa.eu/gender-eu-research-and-innovation_en].

Line 472, citation #54: Medical Science Sex and Gender Equity. About MESSAGE 2023 [Available from: https://www.messageproject.co.uk/about/].

Line 474, citation #55: World Professional Association of Transgender Health. Methodology for the Development of SOC8 2022 [Available from: https://www.wpath.org/soc8/Methodology].

Line 476, citation #56: Murad MH. Clinical Practice Guidelines: A Primer on Development and Dissemination. Mayo Clin Proc. 2017;92(3):423-33.

Line 478, citation #57: Hassan R, Riehl-Tonn VJ, Dumanski SM, Lyons KJ, Ahmed SB. Female sex-specific and -predominant cardiovascular risk factors and heart failure practice guidelines. Am Heart J. 2022;247:63-7.

Line 481, citation #58: Fahmawi S, Schinke C, Thanendrarajan S, Zangari M, Shaughnessy JD, Jr., Zhan F, et al. Under-representation of black patients with multiple myeloma in studies supporting International Myeloma Working Group guidelines. J Cancer Policy. 2023;37:100433.

Line 485, citation #59: Nazer L, Abusara A, Aloran B, Szakmany T, Nabulsi H, Petushkov A, et al. Patient diversity and author representation in clinical studies supporting the Surviving Sepsis Campaign guidelines for management of sepsis and septic shock 2021: a systematic review of citations. BMC Infectious Diseases. 2023;23(1):751.

Line 489, citation #60: Lubarsky R, Ambinder D, Barnett J, Choudhury M, Saji A, Fishman AI, et al. Prostate Cancer: Under-representation of African American Men in Research Studies Used in the Latest NCCN Guidelines. Urology. 2023;180:28-34.

Line 492, citation #61: Zuberi SA, Burdine L, Dong J, Feuerstein JD. Representation of Racial Minorities in the United States Colonoscopy Surveillance Interval Guidelines. J Clin Gastroenterol. 2023.

Line 495, citation #62: Williamson TJ, Battaglia PJ, Gliedt JA, Spector AL, Williams JS. Trials Informing Back Pain Guidelines Underreport Key Sociodemographic Data. J Health Care Poor Underserved. 2023;34(1):357-76.

Reviewer #1: 

I really appreciate this unbiased analysis of the literature from which WPATH have drawn their latest standards of care. I have a number of suggested revisions to maximise its utility to the published literature. Some of these are minor in nature in a described in comments in the PDF attached.

PDF Comments:

C1. Characterize is a broad term. Though the abstract is aiming to be brief, understanding whether aims were around quality/direction etc would the reader.

R1. We thank the Reviewer for this comment. As we aimed to describe the literature (e.g., participant demographics, study design) cited in WPATH SOC 8, we changed the wording from “characterize” to “describe” in the abstract on line 29 of page 2:

“Thus, we aimed to describe the literature informing care provision of TGD individuals.” 

C2. This conclusion links less well to the results, as number of longitudinal studies was not specified. That would be helpful to know in the results to link this, and whether this changed over time.

R2. Thank you for this comment. To determine the proportion of longitudinal studies cited in WPATH SOC 8, we have combined all prospective and retrospective studies, randomized controlled trials, case control and pre-post studies. The abstract has been updated to include this result on lines 39 to 40 of page 2: 

“Almost half of studies (45%) were cross-sectional, and approximately a third were longitudinal in nature (37%).”

C3. Was quality evaluation a factor? "Exploring" hormone therapy guidelines seems vague - especially if only looking at primary studies. Could more detail be provided here? As it’s such a topic of huge interest right now. I think specifying you comment on hierarchy of studies but not full analysis if their rigour e.g. matching/blinding etc.

R3. We thank the Reviewer for this comment. We aimed to describe the literature informing TGD health care rather than evaluate the quality of studies cited. Moreover, we repeated our analysis of chapter 12, titled Hormone Therapy, given the recent heightened focus on this area of TGD health care. As such, we have changed the word “exploring” to “describe” and updated the objective of this study on page 5, lines 82 to 86: 

“Our primary objective was to describe the participant demographics and study design of the literature cited in WPATH SOC 8 informing the care of TGD persons. Our secondary objective was to specifically describe the literature informing gender-affirming hormone therapy guidelines, given recent heightened focus on this area of TGD healthcare.”

C4. I am concerned about the quality this might provide. Is it possible to verify with a native speaker?

R4. We thank the Reviewer for this comment. We have updated lines 99 to 101 on page 6 to include a reference that details the accuracy of Google Translate: 

“Articles in languages other than English were translated to English using Google Translate (1), which has been previously validated in systematic reviews (2).”

C5. Given known high proportion of individuals classing themselves as having a disability, and also the high overlap with neurodivergent traits, it would be very helpful to include where this was specified.

R5. We thank the Reviewer for this comment. We have updated the methods on page 6, lines 109 to 110: 

“Data was also extracted on whether studies reported disability and neurodivergent status of participants (3, 4).”

Data on whether studies report disability and neurodivergent status, can now be found in Table 1, Table 2, in the Supplementary File and on page 7, lines 141-142:

“A minority of studies included disability (22%) or neurodivergent (5%) status of participants.”

and for the “Hormone Therapy” chapter on page 13, lines 197-198:

“Approximately a third of studies reported disability status (32%), and 6% reported neurodiversity status”

C6. I applaud the authors for this inclusion. 

R6. We thank the Reviewer for this comment for our statement on lines 213 to 219: 

“Underrepresentation of older adults in clinical research is widespread across virtually all fields (26-32) with resulting care guidelines often relying on studies with limited older participants, potentially reducing their applicability in this age group (33-35). A recent review (36) outlined common barriers and potential solutions to increase participation of older adults in clinical research, which coupled with TGD-inclusive practices (37, 38) could lead to a greater evidence base in the health care of older adults.”

C7. I think this would be stronger if grounded in the results section. Specifically commenting on number of 'longitudinal studies' (I think you would class prospective and RCT) together, and how this changes over time. Were there any long term cohorts cited at all? E.g. ENIGI

R7. Thank you for this comment. We conducted a similar analysis to the one investigating temporal patterns in the gender composition of study populations, and have included the results as Figure 3. For this analysis we compared temporal patterns in 4 study designs: 1) cross sectional studies, 2) longitudinal studies (includes prospective and retrospective cohort studies, randomized controlled trials, case control, and pre-post studies), 3) qualitative studies, 4) mixed methods studies. Overall, we found that there was a significant decrease in longitudinal and increase in qualitative studies from 1969 to 2023. Cross-sectional and mixed method studies demonstrated no significant change in trends. Please see pages 12 and 13, lines 172-77, for the updated results.

“From 1969 to 2023, we observed a change in the composition of study designs. Longitudinal studies were initially most common, with a temporal increase in cross sectional, qualitative and mixed method studies (Fig 3A). Moreover, the proportion of longitudinal studies decreased (p=0.04) while the proportion of qualitative studies increased (p=0.03; Fig 3B). The proportions of cross sectional (p=0.09) or mixed-methods (p=0.80) studies did not change over the same time period.”

Please see below for Figure 3A and 3B, and the included figure legend on pages 13, lines 178 to 184: 

Fig 3. Temporal patterns in study designs among studies cited in the WPATH SOC 8. A) Absolute number of studies published and study designs as a function of time. B) Design of studies over time expressed as perce

---

## [Decision Letter · Decision Letter 1]

1 Aug 2024

PONE-D-24-13832R1Characterization of the literature informing health care of transgender and gender-diverse persons: A bibliometric analysisPLOS ONE

Dear Dr. Ahmed,

 Thank you for your thoughtful responses to our reviewer's comments. One reviewer suggests some minor changes to add additional context to your article and improve reader interpretation and comprehension. Please take time consider and respond to these comments before submitting what I anticipate will be the final version of your manuscript. Excellent work.

Kind regards,

Christina M. Roberts, M.D., M.P.H.

Academic Editor

PLOS ONE

Journal Requirements:

Reviewers' comments:

Reviewer's Responses to Questions

**Comments to the Author**

1. If the authors have adequately addressed your comments raised in a previous round of review and you feel that this manuscript is now acceptable for publication, you may indicate that here to bypass the “Comments to the Author” section, enter your conflict of interest statement in the “Confidential to Editor” section, and submit your "Accept" recommendation.

Reviewer #1: (No Response)

Reviewer #2: All comments have been addressed

2. Is the manuscript technically sound, and do the data support the conclusions?

Reviewer #1: Yes

Reviewer #2: Yes

3. Has the statistical analysis been performed appropriately and rigorously? 

Reviewer #1: Yes

Reviewer #2: Yes

4. Have the authors made all data underlying the findings in their manuscript fully available?

Reviewer #1: Yes

Reviewer #2: Yes

5. Is the manuscript presented in an intelligible fashion and written in standard English?

Reviewer #1: Yes

Reviewer #2: Yes

6. Review Comments to the Author

Reviewer #1: Many thanks to the authors for addressing previous comments.

I have some further comments around the dicussion which once addressed I feel will make this paper suitable for publication.

1. The authors should acknowledge in limitations the use of Google Translate for studies not in English and provide the N number here.

2. The authors have helpfully commented on inclusion of neurodiverse participants in the Results, this deserves a few lines in the discussion, in the same way as for older adults, as it is a significant proportion of the gender diverse population.

3. The authors have given the proportion of longitudinal studies in the results and commented on this, which is helpful. I note from the results that the number of qualitative studies has also increased which is important as it suggests as it suggests we MAY be better evaluating patient experience in studies - this deserves a comment in the discussion.

4. In line 255 the authors discuss RCT as high quality evidence. It is worth a comment that this may be ethically challenging in this space, given the public discourse around the Cass review in the UK.

Reviewer #2: Thank you for your revision of this article, it will add significantly to the literature on care of TGD populations and keep up the great work!

7. PLOS authors have the option to publish the peer review history of their article (what does this mean?). If published, this will include your full peer review and any attached files.

Reviewer #1: No

Reviewer #2: **Yes: **Michael I. Kruse

---

## [Author Response · Author response to Decision Letter 1]

3 Aug 2024

August 3, 2024

Emily Chenette, PhD 

Editor-in-Chief 

PLOS ONE

Manuscript ID#: PONE-D-24-13832.R2

Title: Characterization of the literature informing health care of transgender and gender-diverse persons: A bibliometric analysis

Thank you for allowing us the opportunity to resubmit our manuscript, based on the thorough review and thoughtful comments provided by the Reviewers. We feel that we have been able to satisfactorily address all the comments raised, which has further enhanced the quality of the manuscript. We have provided an itemized summary of the changes made to the paper below with the Reviewers’ comments provided in bold, followed by our responses. We have provided a tracked changed version of the manuscript, as well as a version without track changes.

Reviewer #1: 

C1. The authors should acknowledge in limitations the use of Google Translate for studies not in English and provide the N number here.

R1. We thank the Reviewer for this comment. We have included the total number of non-English studies that were translated on page 6, lines 99 – 101:

“Articles in languages other than English (n=5) were translated to English using Google Translate (1), which has been previously validated in systematic reviews (2).”

We have also acknowledged this as a limitation of our study on page 21, lines 308 – 313: 

“Moreover, articles in languages other than English were translated using Google Translate (1) into English, rather than having a native language speaker review the studies, potentially impacting the accuracy of our review. Nevertheless, a previous study has validated the use of this translation tool in systematic reviews, demonstrating an 85-97% accuracy when compared to native language speakers (2).”

C2. The authors have helpfully commented on inclusion of neurodiverse participants in the Results, this deserves a few lines in the discussion, in the same way as for older adults, as it is a significant proportion of the gender diverse population.

R2. We appreciate the Reviewer’s comment. We have updated lines 243 – 251 on pages 18 and 19 to include a discussion of neurodivergent study participants: 

“Furthermore, very few studies reported participant disability or neurodiversity status. Previous studies have suggested increased rates of autism, as well as other neurodevelopmental and psychiatric diagnoses in TGD individuals relative to cisgender individuals (3, 4), and thus appropriate representation in study participants is crucial to tailor care meeting the needs of TGD populations. A recent review (5) outlined common barriers and potential solutions to increase participation of older adults in clinical research, which coupled with TGD-inclusive practices (6, 7), could lead to a greater evidence base in the health care of TGD persons.”

C3. The authors have given the proportion of longitudinal studies in the results and commented on this, which is helpful. I note from the results that the number of qualitative studies has also increased which is important as it suggests as it suggests we MAY be better evaluating patient experience in studies - this deserves a comment in the discussion.

R3. We thank the Reviewer for this comment. We have included a discussion of the increase in the number of qualitative studies on page 19, lines 258 – 260: 

“However, the significant increase in the proportion of qualitative studies is encouraging as it may reflect a shift towards a greater number of studies examining TGD individuals’ experiences.” 

R4. In line 255 the authors discuss RCT as high-quality evidence. It is worth a comment that this may be ethically challenging in this space, given the public discourse around the Cass review in the UK.

C4. We appreciate the Reviewer’s comment. We have updated lines 265 – 267 on page 19 with the following: 

“Of note, RCTs are not always appropriate or ethical, particularly in the absence of clinical equipoise, and observational studies may be more reflective of “real world” data (8).

Reviewer #2: 

C1. Thank you for your revision of this article, it will add significantly to the literature on care of TGD populations and keep up the great work!

R1. We thank the Reviewer and greatly appreciate their comment.

References

1. Google. Google Translate 2006 [Available from: https://translate.google.ca].

2. Jackson JL, Kuriyama A, Anton A, Choi A, Fournier J-P, Geier A-K, et al. The Accuracy of Google Translate for Abstracting Data From Non–English-Language Trials for Systematic Reviews. Annals of Internal Medicine. 2019;171(9):677-9.

3.Warrier V, Greenberg DM, Weir E, Buckingham C, Smith P, Lai MC, et al. Elevated rates of autism, other neurodevelopmental and psychiatric diagnoses, and autistic traits in transgender and gender-diverse individuals. Nat Commun. 2020;11(1):3959.

4. Kallitsounaki A, Williams DM. Autism Spectrum Disorder and Gender Dysphoria/Incongruence. A systematic Literature Review and Meta-Analysis. J Autism Dev Disord. 2023;53(8):3103-17.

5. Forsat ND, Palmowski A, Palmowski Y, Boers M, Buttgereit F. Recruitment and Retention of Older People in Clinical Research: A Systematic Literature Review. J Am Geriatr Soc. 2020;68(12):2955-63.

6. Rytz CL, Beach LB, Saad N, Dumanski SM, Collister D, Newbert AM, et al. Improving the inclusion of transgender and nonbinary individuals in the planning, completion, and mobilization of cardiovascular research. American Journal of Physiology-Heart and Circulatory Physiology. 2023;324(3):H366-H72.

7. Owen-Smith AA, Woodyatt C, Sineath RC, Hunkeler EM, Barnwell T, Graham A, et al. Perceptions of Barriers to and Facilitators of Participation in Health Research Among Transgender People. Transgend Health. 2016;1(1):187-96.

8. Toews I, Anglemyer A, Nyirenda JL, Alsaid D, Balduzzi S, Grummich K, et al. Healthcare outcomes assessed with observational study designs compared with those assessed in randomized trials: a meta-epidemiological study. Cochrane Database Syst Rev. 2024;1(1):Mr000034.

---

## [Editor Report · Decision Letter 2]

6 Aug 2024

Characterization of the literature informing health care of transgender and gender-diverse persons: A bibliometric analysis

PONE-D-24-13832R2

Dear Dr. Ahmed,

We’re pleased to inform you that your manuscript has been judged scientifically suitable for publication and will be formally accepted for publication once it meets all outstanding technical requirements.

Kind regards,

Christina M. Roberts, M.D., M.P.H.

Academic Editor

PLOS ONE

Great job!

---

## [Editor Report · Acceptance letter]

8 Aug 2024

PONE-D-24-13832R2 

PLOS ONE

Dear Dr. Ahmed, 

I'm pleased to inform you that your manuscript has been deemed suitable for publication in PLOS ONE. Congratulations! Your manuscript is now being handed over to our production team.

Kind regards, 

on behalf of

Dr. Christina M. Roberts 

Academic Editor

PLOS ONE